# Assessment of Bleeding Risk in Hospitalized COVID-19 Patients: A Tertiary Hospital Experience during the Pandemic in a Predominant Minority Population—Bleeding Risk Factors in COVID-19 Patients

**DOI:** 10.3390/jcm11102754

**Published:** 2022-05-13

**Authors:** Hesham Abowali, Antoinette Pacifico, Burak Erdinc, Karim Elkholy, Umida Burkhanova, Tarilate Aroriode, Althea Watson, Muhammad Faizan Ahmed, Ese Uwagbale, Nathan Visweshwar, Michael Jaglal

**Affiliations:** 1One Brooklyn Health, Brookdale Hospital, Brooklyn, NY 11212, USA; habowali@bhmcny.org (H.A.); uburkhanova@bhmcny.org (U.B.); euwagbale@bhmcny.org (E.U.); 2Baystate Medical Center, Springfield, MA 01199, USA; antoinette.pacifico@baystatehealth.org; 3SUNY Downstate Medical Center, Brooklyn, NY 11203, USA; burak.erdinc@downstate.edu (B.E.); muhammad.ahmed@downstate.edu (M.F.A.); 4Jefferson-Einstein Healthcare Network, Philadelphia, PA 19141, USA; elkholyk@einstein.edu; 5Rochester General Hospital, Rochester, NY 14621, USA; tarilate.aroriode@rochesterregional.org; 6Lincoln Hospital, Bronx, NY 10451, USA; watsonann2021@gmail.com; 7Morsani College of Medicine, University of South Florida, Tampa, FL 33612, USA; nviswesh@health.usf.edu; 8Moffitt Cancer Center, 12902 Magnolia Drive, Tampa, FL 33612, USA

**Keywords:** infection, hemorrhage, thrombosis, SARS-CoV-2, blood transfusion

## Abstract

Introduction: In the wake of the SARS-CoV-2 (COVID-19) pandemic, our world has faced multiple challenges. Infection with this virus has commonly been associated with thrombotic events. However, little is known about bleeding risk and anticoagulation therapy. This study aims to determine factors that are associated with increased risk of bleeding in COVID-19 patients. Methods: A retrospective cohort study was conducted using the records of COVID-19 patients admitted during the COVID-19 pandemic from March 2020 through May 2020. Using patient charts, investigators manually collected data regarding patient characteristics and bleeding. Patients were included in the analysis if they had a confirmed COVID-19 PCR test, were older than 18 years of age and were admitted to the hospital. Patients who were pregnant or had incomplete charts were excluded from the study. ANOVA and logistic regression were used to determine the statistical significance of the data using SPSS version 27. Results: A total of 651 patients were included in the analysis out of 685 patients located in the database of COVID-19 infected patients during that time frame. The general characteristics of the patients were as follows: 54.2% were males; females 45.8% ages ranged from 28 to 83 years old (median age = 66 years old). There were 31 patients (4.9%) who required more than 1 unit of packed red blood cell (PRBC). A total of 16 (2.85%) patients had a documented gastrointestinal bleed (GIB), of which 8 received a total of 29 units of PRBC transfusions. The HAS-BLED score (without alcohol/drug due to inadequate charting) is calculated for patients who had a documented GI bleed and who received more than one unit of PRBC. It was noted that the higher the HAS-BLED score the greater the likelihood of having a GI bleed (*p* < 0.001). The HAS-BLED score (not including alcohol/drug) was also predictive for patients who received more than one unit of PRBC during their hospital stay (*p* < 0.001). Discussion: Using the HAS-BLED score without alcohol/drugs, patients with COVID-19 can be stratified in regard to their risk of GI bleeding and their risk of transfusion while in the hospital. When administering anticoagulation therapy, cautious monitoring should be carried out. Decisions regarding anticoagulant therapy should be based on individual patient characteristics.

## 1. Introduction

The severe acute respiratory syndrome coronavirus 2, SARS-CoV-2 (COVID-19) pandemic has been one of the most challenging diseases in the 21st century and there are multiple complications that are being recognized through ongoing research. COVID-19 has commonly been associated with acute thrombotic events, but less information is evident regarding the risk stratification of bleeding in patients receiving anticoagulation. The role of anticoagulation in COVID-19 is currently being evaluated in multiple studies, as well as clinical trials assessing the prognosis of patients receiving such treatment. The pathogenesis of the hypercoagulable state in COVID-19 is not completely understood. Suggested mechanisms of injury include direct endothelial injury by the virus, immobilization, increased inflammation, and elevation in coagulation factors [1,2,3,4,5].

Multiple studies have reported the benefits and risks of anticoagulation therapy in reducing hospital mortality in patients with COVID-19. Nadkarni et al. found a 50% reduced risk of in-hospital mortality and a slightly increased risk of bleeding in patients who were on therapeutic anticoagulation in their cohort [6]. Both therapeutic and prophylactic doses of anticoagulation therapy were associated with increased in-hospital survival and diminished needs for intubation and mechanical ventilation. In contrast, Musoke et al. reported an increased risk of bleeding and higher mortality with therapeutic anticoagulation in hospitalized COVID-19 patients [7]. Kessler et al. observed a two-fold increase in the incidence of bleeding in patients with COVID-19 who received therapeutic doses of anticoagulation [8]. This study recommended intensive use of thromboprophylaxis in hospitalized patients with COVID-19 and de-escalation of anticoagulant therapy after 10 to 14 days to decrease the risk of bleeding complications in patients with a favorable clinical course.

In contrast to the well-documented hypercoagulable state, bleeding complications of COVID-19 are less common, yet remain high and are mostly seen in critically ill patients [9]. A recent multicenter cohort study with 400 patients by Al-Samkari et al. reported an overall thrombotic complication rate of 9.5% and a major bleeding rate of 4.8% in their patient population [10]. The team observed a higher incidence of bleeding in critically ill versus non-critically ill patients, 7.6% and 3.1%, respectively.

While there is a vast amount of literature that has been published over the last year regarding thrombotic events in COVID-19 patients, there is less literature that focuses on the risk of bleeding. Our study aim is to determine if the HAS-BLED score (H-hypertension, A-abnormal renal and liver functions, S-stroke, B-bleeding, L-labile INR, E-elderly, D-drugs or alcohol), without including drugs/alcohol, provides an efficient risk stratification methodology in assessing patients with COVID-19 infection who may be at a high risk of bleeding. Moreover, another study aim is to evaluate if there is a correlation between specific risk factors and bleeding in COVID-19 patients since conflicting evidence has been reported.

## 2. Methods

This project was generated from an IRB protocol number 20–32 approved by BUHMC Research and Clinical Projects Committee (RCPC/IRB). The initiative was undertaken by the Department of Medicine of the Brookdale Hospital University Medical Center (BUHMC), Division of Hematology/Oncology, at BUHMC. The analysis regarding COVID-19 bleeding risk consisted of data collection and analysis of patients admitted to Brookdale Hospital during the time of 14 March 2020 and 1 May 2020, with positive COVID-19 PCR tests during the first peak of the COVID-19 pandemic.

### 2.1. Data Collection and Analysis

A retrospective cohort analysis was performed. Adult patients who had a positive COVID-19 PCR test between 14 March 2020 and 1 May 2020, were eligible. During the study period, patients were selected according to the following inclusion and exclusion criteria.

Inclusion criteria were patients more than 18 years old who had a positive SARS-CoV-2 PCR test and were admitted to the hospital for medical management. Patients were excluded from the study if they were less than 18 years of age, pregnant females or those who had incomplete charting or documentation.

The primary outcome of this study is to analyze whether the HAS-BLED score (not including alcohol/drugs) correlates with patients’ bleeding events. Bleeding events were defined by one of the following: (i) Gastrointestinal (GI) bleeding in the patients, (ii) patients requiring blood transfusion of more than 1 unit of packed red blood cell (PRBC). The use of more than one unit of PRBC was used as a surrogate since patients’ bleeding was not documented very well in some charts due to the pandemic. The assessment of intracranial bleeding was not routinely assessed for unless clinically indicated. None of the patients were found to have documented intracranial bleeding diagnosis although routine intracranial scanning was not performed in this patient population. Data were extracted manually by investigators from individual patient charts. Other factors gathered included comorbidities and anticoagulation used in patient settings in relation to (i) documented GI bleeds and (ii) patients who received more than 1 unit of PRBC. Patient charts were obtained from a file prepared by the hospital’s department of medicine for patients who were admitted to the hospital with confirmed COVID-19 infection. The charts were reviewed carefully by the investigators. All data obtained were subsequently coded onto a master sheet using a Microsoft office excel spreadsheet (Version 2016, Microsoft Corporation, Redmond, WA, USA). A separate investigator performed weekly data monitoring, and any discrepancies between the patient chart and the master sheet were reviewed and corrected.

### 2.2. Ethical Considerations

This project was also submitted and approved by the local institutional review board (BUHMC RCPC/IRB) for retrospective data analysis. Confidentiality of information was maintained, and all data were collected and de-identified. Data files were secured with coded file access that was only made accessible to investigators.

### 2.3. Statistical Analysis

Basic characteristics of the patients were obtained, and their median and interquartile ranges were calculated as continuous variables. For categorical values, percentages were used. Regarding the primary endpoint, HAS-BLED score minus alcohol/drugs was compared in patients with GI bleeding and patients who received more than 1 unit of PRBC with patients who did not have any evidence of bleeding using analysis of variance with ad hoc analysis. Logistic regression was used to analyze the correlation between bleeding events and other comorbidities and anticoagulation provided in the hospital. All data were analyzed using IBM SPSS Statistics for Windows, Version 27.0 (IBM Corp. Released 2017. Armonk, NY, USA). Significance was determined as *p* < 0.05.

## 3. Results

A total of 685 patients were located on the directory in EPIC during the selected period. A total of 34 patients were excluded from the study. After a thorough review, 18 patients were not admitted to the hospital, five patients died either before admission or within 24 h after admission orders, seven patients were pregnant, two patients were under 18 years old, and two patients were admitted for other issues and were COVID-19 negative by PCR testing. The 34 patients listed above were excluded; hence, a total of 651 COVID-19 admissions were included in the present analysis. The general features of the patients were as follows: 54.2% were males; 45.8% were females; age ranged from 28 to 83 years old (median = 66 years old) (Table 1).

Of the 651 patients in the analysis, 44 (6.76%) had a transfusion of at least 1 unit of packed red blood cells (PRBC), while 31 (4.8%) received more than 1 unit of PRBC. A total of 16 (2.85%) patients had a documented gastrointestinal bleed (GIB), of whom 8 received a total of 29 units of PRBC transfusions.

The analysis of HAS-BLED score (not including alcohol/drug due to deficient charting), with patients who had a documented GI bleed and those who received more than 1 unit of PRBC noted that the higher the HAS-BLED score the higher the chance of having a GI bleed (*p* < 0.0001). A higher HAS-BLED score (not including alcohol/drug) was associated with patients who received more than one unit of PRBC during their hospital stay (*p* < 0.0001).

While analyzing the secondary outcomes to view the effect of co-morbidities and their relation to patients who had GI bleeding events. It was noted that patients with a history of congestive heart failure (CHF) had a higher likelihood to develop such an event (<0.001). The higher rate of bleeding in the CHF patients may be due to the fact that 52.9% of patients with CHF were on anti-platelets in comparison to 20.8% of patients with no CHF history. Moreover, those patients who had coronary artery disease (CAD) were statistically more likely to have a GI bleed (*p* = 0.009). The rest of the comorbidities did not show any statistical significance in relation to GI bleed, diabetes (DM) (*p* = 0.314), cancer (*p* = 0.725), auto-immune disease (*p* = 0.998), and obesity (*p* = 0.684) (Table 2).

Regarding patients’ anticoagulation treatment in the hospital, it was confined to lovenox, except for patients with creatinine clearance < 30 mL/min, the anti-coagulation used was unfractionated heparin. Patients who were on full dose anticoagulation and prophylactic anticoagulation were compared; GI bleeding events were not statistically significant between groups (*p =* 0.615). Moreover, there was no statistical difference between both groups about patients requiring blood transfusions (*p =* 0.997).

In patients who required more than 1 unit of PRBC, it was noted that DM had a statistically significant decrease in events when compared to non-diabetics (*p =* 0.022). Obesity (*p =* 0.11), CHF (*p =* 0.929), cancer (*p =* 0.997), CAD (*p =* 0.78) and auto-immune diseases (*p =* 0.998) did not show any significant correlation with the patients requiring more than 1 unit of PRBC transfusion.

Patients who received at least one unit of PRBC transfusion were more likely to take an anti-platelet or NSAID as a home medication (22/212 vs. 22/437, *p =* 0.011).

## 4. Discussion

In our study, we focused on investigating the correlation between the presence of the COVID-19 virus and risk factors for bleeding in treated patients. There have been multiple studies that suggest increased thrombosis in such patients, but the mechanism by which this occurs remains unclear [1,2,3,4,5]. Many patients with COVID-19 infection have required the initiation of full-dose anticoagulation given the increased risk of thrombosis [8]. The likelihood of bleeding in such patients is not fully understood, as it could be due to the presence of anticoagulation therapy, the virus itself or from increased risk of bleeding in certain patient populations.

Of the 651 patients in our study, 461 (70.9%) were on prophylactic anticoagulation, 128 (19.7%) were on full-dose anticoagulation, and 61 (9.4%) were not on any anticoagulation therapy. The median peak d-dimer was found to be 903 ng/mL (1st/3rd IQ: 505–4953 ng/mL) and was based on 303 patients who had this data in their medical records. Because our data collection occurred at the beginning of the pandemic, d-dimer values were not reported for every patient and the importance of this lab value was not understood at that time. The HAS-BLED score, without including alcohol/drugs, was used to assess each patient’s risk of bleeding in our study. This score is used to determine a patient’s bleeding risk with atrial fibrillation. This score includes a point system that stratifies patients into low, medium, and high risk of bleeding based on factors such as age, medical disorders, and prior episodes of bleeding [11]. Since the d-dimer values for our patients were high, meaning that they could be at increased risk of developing thromboembolism, it was felt that the benefit of anticoagulation therapy outweighed the risks of bleeding. Of note, alcohol/drugs were not factored into the HAS-BLED scores in our study since this information was not collected for all patients.

In our study, we categorized bleeding as significant by measuring the need for transfusion of 1 unit of PRBC. The International Society on Thrombosis and Hemostasis (ISTH) demonstrated that transfusion of one PRBC is equivalent to significant bleeding [12]. We chose this parameter as a secondary measure of bleeding along with GI bleeding. According to the ISTH, receiving one PRBC means the individual has sustained around a 3% change in hematocrit, corresponding to around 500 cc of volume loss. Since the ISTH considers this a significant drop, we used this metric to categorize bleeding in COVID-19 positive patients while they were in the hospital [12,13]. Documentation errors in bleeding that could have occurred during a patient’s stay in the hospital can be accounted for by this measure. By using the need for PRBC transfusion, charting bias is eliminated. Furthermore, the study included only patients admitted for medical management. We also reported the bleeding complications of 651 COVID-19 patients and investigated the relationship between bleeding risk and aspirin, antiplatelet or NSAID use. A total of 44 out of 651 patients (6.76%) received at least one unit of packed red blood cells. Eight patients who were diagnosed with gastrointestinal bleed received a total of 29 units of PRBCs. It is unclear if COVID-19 can cause direct damage to the GI epithelium resulting in ulceration and bleeding via ACE-2 (acetylcholine esterase-2) receptors [14]. There are only a few case reports and case series of patients who had GI bleed after they were diagnosed with COVID-19 in the current literature [15,16,17]. Therefore, the relationship between COVID-19 and GI bleed is unclear and there is not enough data to support causality. Holzwanger et al. published a case series of 11 patients who presented with a lower GI bleed and various severity of COVID-19 [18]. However, 8 out of 11 patients were already on anticoagulation therapy at the time of GI bleed. A case report by Carnevale et al. discussed the histopathological findings in the small bowel of a 40-year-old woman infected with COVID-19. Laboratory evidence of anemia and positive fecal occult blood test in this patient lead to diagnostic colonoscopy. Biopsies were taken from both the mucosa and submucosa, specifically from two ulcerations found at the ileocecal valve. These samples showed the presence of T-lymphocytes, multifocal vasculitis and obliterating arteriolitis. Immunohistochemical staining detected the presence of viral particles in the cytoplasm of endothelial cells. The findings from this study suggest that COVID-19 may damage the GI tract in patients due to a combination of hyperinflammation, hypercoagulability, and direct endothelial damage [19].

Additionally, we found that patients with a higher HAS-BLED (excluding alcohol/drug) score had a significantly higher chance of GI bleeding during the hospital course (*p* < 0.0001). A HAS-BLED score without the input of alcohol or drugs is not a validated scoring system. However, the lack of documentation in the charts for alcohol or drug use limited our ability to include this factor; hence, this is an additional limitation of this study. We suggest that this scoring system can be utilized to assess the bleeding risk in COVID-19 patients. This score can be calculated prior to starting patients on anticoagulation therapy for COVID-19-related coagulopathy without a confirmed indication such as deep vein thrombosis (DVT), pulmonary embolism (PE), acute myocardial infarction (MI) or acute limb ischemia. This will help establish a baseline bleeding risk allowing clinicians to make more informed decisions regarding which type and dose of anticoagulant to use. Since reversal agents for each anticoagulant are different, a patient with a higher risk of bleeding may benefit from an anticoagulant that has a short half-life or a readily available reversal agent. A patient with a lower risk of bleeding could have more anticoagulation options. A report by Thachil et al. discussed some of the research that has been conducted so far regarding heparin and low molecular weight heparin (LMWH) for the treatment of thrombosis in COVID-19 patients. While these two agents are the most popular and have the most research in regard to dosage and indication, several studies are still ongoing regarding the biomarkers of coagulation and anticoagulation in patients with COVID-19. Current guidelines suggest treating patients with heparin or LMWH and recommend increasing dosage based on disease severity [20].

In our analysis, a greater risk of bleeding was seen in our patient population that had CHF. Patients with CHF were found to have an increased risk of bleeding. We suggest further research to find out if CHF patients consistently experience increased GI bleeding with COVID 19 infection. The higher rate of bleeding in the CHF patients may be due to the fact that there was a higher rate of antiplatelet drugs in that patient population.

Even though aspirin (ASA) and NSAIDs (non-steroidal anti-inflammatory drugs) can be used in patients with COVID-19 with the aim of decreasing inflammation and hypercoagulable state, there is no data supporting any mortality benefit [21,22,23]. We found that our patients with COVID-19 who took aspirin, anti-platelets or NSAIDs as a home medication did have a higher risk of bleeding compared to patients who were not taking these medications. Patients who received at least one unit of PRBC transfusion were more likely to take an anti-platelet or NSAID as a home medication (27.623, 22/212 vs. 22/437, *p* = 0.011). Hence, we recommend continuing ASA and NSAIDs if there is an established indication. Further studies that research the possible benefits of the use of ASA and NSAIDs in these patient populations are needed.

In our patient population, which was noted to have a pre-dominant African American population, the bleeding events were high for hospitalized patients with COVID-19 infection. Data from the NHS reported that African American and Asian populations are at higher risk of infection and death in 2020 [24]. However, another study generated from Louisiana reached a conclusion that the black race is not subjected to increased death from COVID-19 [25]. Ibba, et al. add in their review article that even before COVID-19, the incidence of VTE was higher in black and Caucasian races in comparison to the Asian race. Another study conducted in New York City for ICU hospitalized patients with COVID-19 and GI bleeding, showed no difference in regard to race or previous anticoagulation or antiplatelet therapy on the events of GI bleeding; however, it showed a trend towards more bleeding in the African American population after initiation of an anticoagulation protocol [26]. Another study conducted to assess the relationship between bleeding and full-dose anticoagulation reached the conclusion that full-dose anticoagulation was associated with worse outcomes for the patients and increased the incidence of bleeding in a predominantly African American population [7].

A study by Adam Cuker et al. (2021) proposed the most recent guidelines for the administration of anticoagulation for those affected by COVID-19. The study authors recommend prophylactic anticoagulation over intermediate-intensity therapy for patients with COVID-19-related critical illness who are not suspected of or confirmed to have VTE. [27] In addition, the study recommended prophylactic-intensity anticoagulation over intermediate-intensity or therapeutic-intensity anticoagulation for patients with acute illness associated with COVID-19 but who do not have suspected or confirmed VTE. Recently, the authors have expanded their recommendations to suggest that outpatient anticoagulant thromboprophylaxis should not be used in those with COVID-19 who are discharged from the hospital and who do not have suspected or confirmed VTE or another indication for anticoagulation.

There were no significant differences between prophylactic dose anticoagulation and full-dose anticoagulation in regard to bleeding risk in COVID-19 positive patients in our study. The recently reported randomized clinical trials have confirmed this stating that not all patients hospitalized with COVID-19 infection will benefit from higher intensity anticoagulation strategies in preventing thrombosis or bleeding episodes. In the HEP-COVID study with therapeutic LMWH vs. standard AC prophylaxis, the major bleeding rate was 2.4% vs. 2.3% [28]. In the ACTION trial using therapeutic rivaroxaban or LMWH for 30 days vs. standard AC prophylaxis, the major bleeding rate was 3% vs. 1% [29]. In the RAPID trial with therapeutic LMWH or UFH for up to 28 days vs. standard AC prophylaxis, the major bleeding rate was 0.9% vs. 1.7% [30]. In the ATTACC study, using therapeutic UFH or LMWH for up to 14 days vs. standard AC prophylaxis, the major bleeding rate was 1.9% vs. 0.9% [31]. These studies reported that the critically ill patients with COVID-19 did not seem to benefit from the escalation of their anticoagulation. The primary outcome of venous or arterial thrombosis at 30 days was not lower in those randomized to the higher intensity regimen (HR, 1.06; 95% CI, 0.83–1.36) [32]. One of the reasons for the lack of response from anticoagulation therapy in Covid-19 may be distinctive endothelialitis of the pulmonary vasculature with severe endothelial injury associated with the presence of intracellular virus and disrupted cell membranes with widespread thrombosis with microangiopathy [33].

## 5. Conclusions

The number of studies that discuss risk factors for bleeding in COVID-19 infected patients appears to be limited. The HAS-BLED score minus drugs/alcohol might be a tool that can be utilized to risk stratify bleeding risk in patients with COVID-19 infection. It appears that patients who are already at high risk of bleeding should be cautiously monitored if these patients are anticoagulated. More studies are needed to verify this finding.

## Figures and Tables

**Table 1 jcm-11-02754-t001:** Baseline characteristics of 651 patients admitted to Brookdale hospital.

Baseline Characteristic		Median (IQR)/Number (%)
Age		66 (28–83)
Male		353 (54.2%)
Female		298 (45.8%)
Race		
	African American	440 (67.6%)
	White	23 (3.5%)
	Hispanic	94 (14.4%)
	Other	19 (2.9%)
	N/A	75 (11.5%)
Labs		
	D-dimer peak levels	905 (508–4924) ng/mL
	CRP peak levels	8.7 (5.6–21.5) mg/dL
	Troponin peak levels	0.03 (0.012–0.153) ng/mL
	PTT peak levels	31 (28.2–34.6)
	PT peak levels	14.6 (13.5–16.8)
	Ferritin peak levels	582 (280–1000) ng/mL
Significant medical history/medications		
	Anticoagulants Use	24 (3.7%)
	Anti-platelets	101 (15.5%)
	Dual anti-platelets	11 (1.7%)
	History of bleeding disorder	0 (0%)
	History of bleeding	25 (3.8%)
	Diabetes Mellitus	307 (47.2%)
	Hypertension	473 (72.7%)
	Abnormal kidney functions	204 (31.3%)
	Abnormal liver functions	105 (16.1%)
	Cancer history	42 (6.5%)
In-hospital anticoagulation		
	Prophylactic	461 (70.8%)
	Full dose	128 (19.7%)
	None	62 (9.5%)

**Table 2 jcm-11-02754-t002:** Characteristics used to compare GIB patients with non-GIB patients and also used to compare patients who received blood transfusion compared to those who did not receive blood transfusion.

Characteristics Compared	Number of Patients	Percentage	*p*-Value
HAS-BLED score/GI bleed			
0	1/67	1.5%	
1	0/119	0%	
2	3/148	2%	
3	3/158	1.9%	
4	5/106	4.7%	
5	1/43	2.3%	
6	3/9	33.3%	
7	0/1	0%	<0.0001
HAS-BLED score/PRBC transfusion (>1 unit)			
0	2/67	3%	
1	6/119	5%	
2	4/148	2.7%	
3	2/158	1.3%	
4	9/106	8.5%	
5	5/43	11.6%	
6	3/9	33.3%	
7	0/1	0%	<0.0001
Comorbid conditions w GI bleed/Comorbid conditions			
Congestive heart failure (CHF)	8/102	7.8%	
Diabetes Mellitus (DM)	10/307	3.2%	
Cancer	1/42	2.4%	
Autoimmune disease	0/29	0%	
Coronary artery disease (CAD)	3/103	2.9%	
Obesity	6/224	2.7%	
Comorbid condition receiving >1 unit of PRBC/total Comorbid condition			
Congestive heart failure (CHF)	5/102	4.9%	
Diabetes Mellitus (DM)	20/307	6.5%	
Cancer	0/42	0%	
Autoimmune disease	2/29	6.9%	
Coronary artery disease (CAD)	5/103	4.9%	
Obesity	10/224	4.5%	
Receiving transfusion while using ASA/NSAIDs/anti-platelets	22/212	10.3%	
Receiving transfusion while not using ASA/NSAIDs/anti-platelets	22/437	5%	0.011
GIB with Use of ASA/NSAIDs/anti-platelets	8/212	3.7%	
GIB without Use of ASA/NSAIDs/anti-platelets	8/438	1.8%	0.133

## Data Availability

Data can be requested from investigators.

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
