# Peer review of "Assessment of Bleeding Risk in Hospitalized COVID-19 Patients: A Tertiary Hospital Experience during the Pandemic in a Predominant Minority Population—Bleeding Risk Factors in COVID-19 Patients"

_jcm, 2022, doi:10.3390/jcm11102754_

Round 1

Reviewer 1 Report

This is an interesting paper about the bleeding risk in COVID19 patients. The paper is clinically meaningful. However, I have a few suggestions which could in my opinion improve the work in its current form:

  • I think "bleeding risk" in the title is not correct because the study does not include all different types of bleeds in the human body. For an example intracranial bleeds which might be associated with COVID19 will not require blood transfusion but are of great clinical importance. This is not discussed in the paper.
  • The authors did not include alcohol and drug use in the HAS BLED score. Would it have an impact on results if included? Please discuss
  • My biggest concern is that abstract and introduction point to anticoagulation as a central study topic. However what kind of anticoagulation is used and in how many patients is not clear. In the result section I find the following statement: "Regarding, patients anti-coagulation treatment in the hospital, both prophylactic and full dose anti-coagulation were associated with having less GI bleed events, in contrast to those who were not on anticoagulation. Most of the patients who had a bleed were not subjected to any anticoagulation in the hospital." Please provide numbers and data in results section, not in discussion. It is contraintuitive that anticoagulation is associated with less bleeding risk. How do you explain this? I think it is of critical importance to present this topic very clear to the reader. 
  • In the results section I find the following statement: "It was noted that patients with history of congestive heart failure (CHF) had a lower likelihood to develop such an event (<0.0001);" However, in the discussion it says "In our analysis, a greater risk of bleeding was seen with our patient population that had CHF. Patients with CHF were found to have an increased risk of bleeding". Please clarify
  • A study flow chart would be helpful to the reader
  • The discussion is in my opinion too long and 

Author Response

First of all, thank you for your review

dividing the responses into points and to try to be thorough

1.) Regarding the bleeding risk title, we added in the methods that assessment of other forms of bleeding such as intracranial bleeds was not assessed in the methods section, given no patient was presenting with concerns as such and also due to limitations of scans done to fulfill such a criteria.

2.) regarding the absence of drug/alcohol use in the study, we also added in the discussion, while discussing HAS-BLED score that it is a limitation of the study, however due to lack of documentations, it was not added to the score and we understand that this may affect the scoring system of HAS-BLED.

3.) Regarding type of anti-coagulation used we stated that it was lovenox strictly, unless patients had Crcl <30 when we used unfractionated heparin on our patients. None was given oral anti-coagulants.

4.) Connecting to the previous point, and apologize for such a miss-statistical use of comparing prophylactic vs full dose anti-coagulation vs patients who had stopped or not used anti-coagulation. As a good portion of patients who were not on anti-coagulation at the end of the study was because they bled before. we ran another statistical analysis comparing those on prophylactic and full dose anti-coagulation and no statistical difference was noted between both. As a matter of fact, GI bleeding episodes and transfused patients were almost the same percentage between both groups. we also added that and removed the previously discussed paragraph to help eliminate any miss-understanding that may result.

5.) Finally, regarding the CHF results, it was actually significantly increased risk of bleeding in the CHF group as the discussion stated and it was fixed in the results. We also analyzed the use of anti-platelets in those patients and was found that patients with CHF and was hospitalized were doubled in percentage comparted to non-CHF patients and that would be the cause of increased bleeding. That was also corrected in the new revision we are about to submit.

In regards to the prolonged discussion we tried to be as thorough as we can to provide maximal amount of information available to the readers.

I really appreciate the careful review and would like to apologize again about the mistakes that was found during the review.

Reviewer 2 Report

Authors well describe the association betweennbleeding risk and clinical bleedings in their retrospective analysis.

yet their cohort is very “old” in the covid literature, data in thrombosis and bleedings have been described Also in most recent cohorts in the world and sementì be different for each viral variant of concern. This aspect is relevant and should be analyzed with data in the manuscript ora a study limitations

Author Response

Thank you for your input.

We recognize that the data was old while submitting the manuscript, but we wanted to analyze a scoring system that would apply to be appropriate in assessing the bleeding risk for the patients. Additionally, in the discussion we tried to include most of the well studied publications that did assess bleeding events in those patients category. 

We also believe that different variants of COVID that is present will provide a limitation of the study but hopefully this study will help us build an insight about developing a scoring system to further categorize COVID patients in need for anti-coagulation.

Again thank you for your careful review.

Round 2

Reviewer 1 Report

Dear Authors,

Unfortunately I do not see enough improvement of the revised manuscript.  I therefore can not recommend to accept the manuscript. 

Author Response

Dear Sir/madam

I kindly responded to every comment you made about the manuscript and your response is that it is not enough. I do not see how can I further improve the manuscript with no input from your side.

Thank you